# Concept-Based Masking: A Patch-Agnostic Defense Against Adversarial Patch Attacks

**Ayushi Mehrotra**
California Institute of Technology
amehrotra@caltech.edu

**Derek Peng**
University of California, Berkeley
derekpeng@berkeley.edu

**Dipkamal Bhusal**
Rochester Institute of Technology
db1702@rit.edu

**Nidhi Rastogi**
Rochester Institute of Technology
nxrvse@rit.edu

## Abstract

Adversarial patch attacks pose a practical threat to deep learning models by forcing targeted misclassifications through localized perturbations, often realized in the physical world. Existing defenses typically assume prior knowledge of patch size or location, limiting their applicability. In this work, we propose a patch-agnostic defense that leverages concept-based explanations to identify and suppress the most influential concept activation vectors, thereby neutralizing patch effects without explicit detection. Evaluated on Imagenette with a ResNet-50, our method achieves higher robust and clean accuracy than the state-of-the-art PatchCleanser, while maintaining strong performance across varying patch sizes and locations. Our results highlight the promise of combining interpretability with robustness and suggest concept-driven defenses as a scalable strategy for securing machine learning models against adversarial patch attacks. Our code is available at https://github.com/ayushimehrotra/concept-masked-defense.

## 1 Introduction

The deployment of machine learning models in real-world applications requires robustness against strategic manipulation by adversaries. A potent and practical threat is the adversarial patch attack, where an attacker modifies a small region of an input image to induce a targeted misclassification at test time [1, 2, 3]. Such attacks can be realized in the physical world by attaching a printed sticker to an object, making them a direct threat to systems like autonomous vehicles and facial recognition.

To counter this threat, several defense mechanisms have been developed to augment existing classification pipelines. Many of these defenses focus on explicitly identifying the adversarial patch in an input image in order to blur, mask, or otherwise neutralize its effect before it is processed by the model [4, 5, 6]. While effective to a degree, a significant limitation of the current state-of-the-art is that these defenses often rely on strict assumptions about the attack, such as a predefined patch size [6]. This leaves models vulnerable to attacks that fall outside these narrow conditions.

In this work, we propose a new defense strategy that is agnostic to the patch's specific characteristics, including its size, shape, and location. Our approach leverages CRAFT [7], a concept-based explanation method that uses recursive non-negative matrix factorization to extract concept activation vectors. We hypothesize that the pixels comprising an adversarial patch are, by their nature, highly influential and will be captured in the most important concepts. Our defense mechanism involves masking the top percentage of the most important concept activation vectors, effectively neutralizing

39th Conference on Neural Information Processing Systems (NeurIPS 2025) Workshop: Reliable ML from Unreliable Data.

the attack without needing to explicitly detect the patch itself. Our experimental results demonstrate that this method not only outperforms the state-of-the-art PatchCleanser [6] in robust accuracy across all attack intensities but also better preserves accuracy on clean images.

## 2 Related Work

Several defenses have been proposed against adversarial patches, though most rely on restrictive assumptions. Minority Reports [8] is able to systematically occlude input regions, but requires the classifying model to be previously trained on occluded images. PatchCleanser [6] masks input regions and aggregates predictions, but requires prior knowledge of patch size, limiting its practicality. Jujutsu [9] leverages self-supervised feature purification but introduces high computational cost and retraining overhead. Watermarking-based defenses protect data at capture time, yet assume control over the input pipeline and cannot handle arbitrary user inputs. SentiNet [10] detects suspicious regions using saliency maps, but remains a detection-only framework and is vulnerable to adaptive attacks. These limitations motivate the need for patch-agnostic defenses that generalize across patch sizes and locations without costly assumptions. To address this gap, we leverage techniques from model interpretability, a paradigm that has proven effective against related threat models, for instance, in detecting various $L_p$-norm attacks [11, 12, 13, 14].

## 3 Methodology

### 3.1 Threat Model

We consider a multi-class black-box image classifier $f : \mathcal{X} \to \mathcal{Y}$ that maps an input image $\mathbf{x} \in \mathbb{R}^d$ to a label $y \in \mathcal{Y}$. An adversarial patch is defined as a localized perturbation applied to an image, producing $\mathbf{x}_p \in \mathbb{R}^d$, such that $f(\mathbf{x}_p) \neq f(\mathbf{x})$. Unlike global perturbations constrained in $\ell_p$ norm, a patch attack allows the adversary to arbitrarily alter a restricted spatial region, with no bound on pixel magnitude. We assume the attacker has complete freedom to place the patch at any location within the image and that the size of the manipulated region is unknown but significantly smaller than the image dimensions.

Our objective is to design a defense $\mathbb{D}(f, \mathbf{x}) \in \mathbb{R}^d$ such that for any adversarially patched input $\mathbf{x}_p$, the defended prediction remains consistent with the original clean image, i.e., $f(\mathbb{D}(f, \mathbf{x}_p)) = f(\mathbf{x})$.

### 3.2 Defense Architecture

Our defense is built upon the hypothesis that adversarial patches introduce spurious, highly influential features that can be identified and suppressed using concept-based explanations. We use the CRAFT framework [7], a state-of-the-art concept-based explanation method, to decompose a model's internal activations into a set of interpretable concept vectors. By quantifying the importance of these concepts, we can isolate the ones most likely associated with the patch and neutralize their impact. We provide visualization in Appendix A.

#### 3.2.1 Step 1: Concept Extraction and Scoring

Our method first leverages the CRAFT pipeline to discover a basis of interpretable concepts for the classifier. We first collect a set of reference images for each class $y_i$. Specifically, we define a class-conditioned dataset

$$\mathcal{C}_i = \{\mathbf{x}_j : f(\mathbf{x}_j) = y_i, 1 \leq j \leq n\}$$

For each $\mathcal{C}_i$, we apply recursive non-negative matrix factorization (NMF) to the intermediate activations produced by the classifier, yielding a set of concept activation vectors (CAVs), denoted by

$$\text{CRAFT}(\mathcal{C}_\rangle) = \mathbf{W}_i = \{\mathbf{w}_{i1}, \mathbf{w}_{i2}, ..., \mathbf{w}_{ik}\}$$

Each CAV $\mathbf{w}_{ij}$ corresponds to a semantically coherent region of the input space.

To determine which concepts are most critical for classification, we employ the Sobol index [15] based importance scoring. This variance-based sensitivity analysis quantifies each concept's contribution to the model's prediction variance. Our core hypothesis is that an adversarial patch will disproportionately activate one or more of these highly-ranked, influential concepts.

### 3.2.2 Step 2: Patch Suppression via Pixel Masking

Given a new test image $\mathbf{x}$ (which may or may not be patched), our defense first identifies the top-$m$ most influential concepts relevant to its predicted class. We then generate spatial activation maps for each of these concepts. To suppress the regions most susceptible to being part of a patch (for a patched image), we apply a spatial blur to the top n% of pixels with the highest activation values within each of these selected maps. Formally, we obtain a new image:

$$\mathbb{D}(f, \mathbf{x}) = \mathbf{x} + \text{Blur}(\bigcup_{j=1}^{m} \text{Top-n\%}(\mathbf{w}_{ij})).$$

The two main hyperparameters of our defense are $m$, the number of top-ranked concept activation vectors considered for patch suppression, and $n$, the percentage of pixels blurred within each concept map. In Section 4.2, we evaluate how the choice of $(m, n)$ creates a trade-off between adversarial robustness and fidelity on clean images.

## 4 Results

### 4.1 Experimental Setup

We evaluate our defense on Imagenette [16], a ten-class subset of ImageNet [17] designed for fast-benchmarking, using a ResNet-50 classifier [18]. We compare our method against the current state-of-the-art defense, PatchCleanser [6]. Note that PatchCleanser requires prior knowledge of the patch size to configure its masking strategy, whereas our defense is patch-size agnostic. To ensure a comparable setup, we modify the number of masks that PatchCleanser generates from 6 masks to 3 masks. We set our defense hyperparameters to $n = 5\%$ (percentage of pixels blurred) and $m = 2$ (number of top-ranked concept activation vectors) based on the hyperparameter tuning, discussed in Section 4.2.

The primary evaluation metric is robust accuracy, defined as the classification accuracy on images that have been confirmed to be successfully attacked (i.e., misclassified by the undefended model).

Table 1: Comparison of robust accuracy under adversarial patch attack of 1%, 2% and 3% patch-size attacks. Clean denotes the accuracy in recovering model prediction on unperturbed samples.

| Defense | Clean | 1% | 2% | 3% |
|---|---|---|---|---|
| Undefended | 0.998 | 0.00 | 0.00 | 0.00 |
| PatchCleanser | 0.969 | 0.922 | 0.912 | 0.903 |
| Our Defense | 0.979 | 0.944 | 0.960 | 0.959 |

Table 4.1 summarizes the performance of our defense against PatchCleanser and an undefended model under patch attacks covering 1%, 2%, and 3% of the image area.

As expected, the undefended classifier's accuracy drops to 0%, demonstrating the effectiveness of the attacks. While PatchCleanser achieves strong recovery, it relies on prior knowledge of the patch-size. Our proposed method consistently outperforms it across all attack intensities. Notably, our defense not only achieves higher robust accuracy but also maintains a higher clean accuracy, indicating that our masking strategy is more targeted and less disruptive to benign images. This highlights the practical advantage of a patch-agnostic, concept-based approach.

### 4.2 Hyperparameter Tuning

We analyzed the sensitivity of our defense to its two main hyperparameters: $n$, the percentage of pixels masked, and $m$, the number of top-ranked concepts used. These experiments reveal a clear trade-off between robustness and clean accuracy.

**Effect of Top $n\%$ Pixels.** With $m$ set as 2, we varied the percentage of masked pixels. Table 4.2 shows that larger values of $n$ yield stronger robustness at the cost of slightly reduced clean accuracy. However, extremely low values of $n$ (e.g., 1% or 2%) sharply degrade robustness, showing insufficiency in neutralizing the patch. We find $n \in [5\%, 10\%]$ offers the best balance between clean and robust accuracy.

Table 2: Robust accuracy of defense with different top $n\%$ (with $m = 2$).

| Top $n\%$ | Clean | 1% | 2% | 3% |
|---|---|---|---|---|
| n = 10% | 0.956 | 0.956 | 0.970 | 0.979 |
| n = 9% | 0.956 | 0.956 | 0.970 | 0.979 |
| n = 8% | 0.968 | 0.955 | 0.972 | 0.975 |
| n = 7% | 0.968 | 0.954 | 0.972 | 0.975 |
| n = 6% | 0.979 | 0.944 | 0.959 | 0.959 |
| n = 5% | 0.979 | 0.944 | 0.960 | 0.959 |
| n = 4% | 0.988 | 0.882 | 0.880 | 0.869 |
| n = 3% | 0.988 | 0.882 | 0.880 | 0.869 |
| n = 2% | 0.996 | 0.619 | 0.572 | 0.521 |
| n = 1% | 0.996 | 0.618 | 0.572 | 0.521 |

**Effect of Number of Concepts** $m$**.** With $n$ fixed at 10%, we varied the number of concepts. Table 4.2 shows that using too few concepts ($m = 1$) reduces robust accuracy, as the patch may activate multiple influential concepts. Conversely, using too many concepts ($m \geq 4$) over-suppresses class-relevant regions, hurting clean accuracy. We find $m = 2$ or $m = 3$ provides the optimal trade-off.

Table 3: Robust accuracy of defense with different numbers of concepts ($n = 10\%$).

| # of Concepts | Clean | 1% | 2% | 3% |
|---|---|---|---|---|
| $m = 1$ | 0.970 | 0.920 | 0.973 | 0.974 |
| $m = 2$ | 0.954 | 0.956 | 0.970 | 0.979 |
| $m = 3$ | 0.937 | 0.949 | 0.970 | 0.971 |
| $m = 4$ | 0.924 | 0.947 | 0.959 | 0.957 |
| $m = 5$ | 0.912 | 0.932 | 0.947 | 0.945 |

## 5 Limitation

A key limitation of our defense is its reliance on the CRAFT framework. Prior research has demonstrated that saliency-based explanation methods can be vulnerable to adaptive adversarial attacks [19]. An adversary specifically aware of our defense could potentially craft a patch that manipulates the concept activations themselves, thereby evading detection. Investigating the resilience of our defense against such adaptive attacks is an important direction for future work. Similarly, the inherent lack of faithfulness of CRAFT to the underlying model can result in concepts that do not perfectly represent the model's internal reasoning. This could our defense to mask pixels adjacent to, but not exactly aligned with, the adversarial patch, potentially reducing its performance.

## 6 Conclusion

In this work, we introduced a patch-agnostic defense against adversarial patch attacks that leverages concept-based explanations to neutralize adversarial attacks. By masking pixels associated with the most influential concepts, our method outperforms the state-of-the-art, PatchCleanser, in both robust and clean accuracy without requiring prior knowledge of the patch's size. Future work will focus on evaluating the defense against adaptive attacks that target the explanation mechanism itself, extending the approach to larger and diverse datasets, and exploring alternative more faithful concept discovery framework.

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

# A  Visualizations of Defense

**1% Patch Size**    **2% Patch Size**    **3% Patch Size**

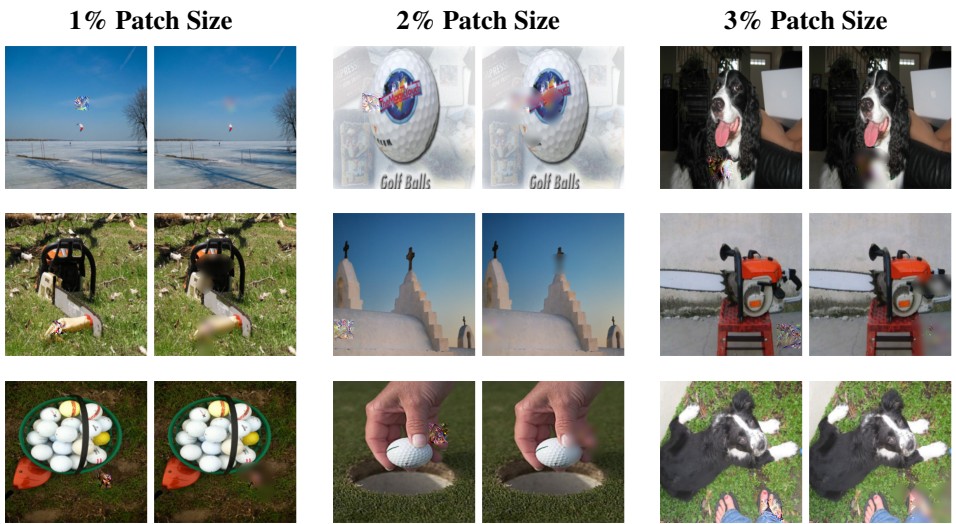

Figure 1: Defense results grouped by patch size (columns) and examples (rows). All images are scaled uniformly for easier comparison.

