# OpenReview forum: "Concept-Based Masking: A Patch-Agnostic Defense Against Adversarial Patch Attacks"
_NeurIPS.cc/2025/Workshop/Reliable_ML — NeurIPS 2025 - Reliable ML Workshop_

### Official Review · Reviewer_vrjF · 2025-09-15
**Promising idea, but evaluation is too limited to support strong claims**

**Rating:** 3
**Confidence:** 3

**Review:**

**Summary:** The paper proposes a method that shields against adversarial patch attacks in image classification. The method identifies parts of the image that may have been adversarially altered and blurs them, reducing the effect that adversarial changes may have on the output. Identifying the regions of interest is performed in a black box manner, using the CRAFT framework. On Imagenette, the method reports higher robust accuracy than PatchCleanser while maintaining clean accuracy.

**Strengths:** The proposed method (unlike previous work) does not require prior knowledge of the size (number of pixels) or the location of the adversarial attack, and is simple enough to implement. Additionally, their method outperforms the state-of-the-art method in robust accuracy (metric that they consider).

**Weaknesses / Limitations:** The work is only tested on Imagenette, a 10-class subset of ImageNet with 100 times less data, and using one classifier (ResNet-50). This does not seem adequate to argue conclusive evidence of the superior performance of the method. Additionally, to compare their method, the authors consider “1%, 2% and 3% patch-size attacks”, which they do not explicitly define. This begs the question as to whether their proposed method is better capable in dealing with some unrealistic attack, which the state of the art fails to capture, but in realistic attacks, their method proves inferior. Finally, the method relies in a black box manner on the CRAFT framework, which suggests that an attack tailored against CRAFT may significantly reduce the method’s effectiveness.

**Suggestions for Authors:** The method is interesting and simple, but the evaluation of it is lacking. Additionally, 1-2 datasets should be considered, as well as different implementations of classifiers. Additional information in regards to the type of attack used in the evaluation is required (are 1%, 2% and 3% patch size attacks what people use to attack image classifiers, or are these the first benchmark to consider?). Finally, the submission would strengthen significantly, should it involve adaptive to CRAFT attacks, and implementations that counteract these attacks.

**Ethics:** No concerns.

---

### Official Review · Reviewer_qo1P · 2025-09-16
**on topic, workshop level paper**

**Rating:** 7
**Confidence:** 5

**Review:**

This paper introduces a patch defense algorithm based on activation vectors. Roughly stated the defense

1. uses an interpretability method to find k concept vectors per class in multi class classification
2. at test time, finds the top m  such concepts per class
3. blurs the top n% of pixels that aligned with the concept
4. inference on the new image

The evaluation results compare favorable to patch cleanser (which is a strong baseline (SOTA?)). The authors note that their results do not need to know the patch size, which is a strong benefit of their algorithm.

Some weaknesses:

-The main algorithm in the paper could be written more explicitly, e.g. explain more precisely step 4 and how the  top n% of pixels with the highest activation values are computed.
-the patch attack algorithm they evaluate on  is not explained very clearly. I checked the code and it is a form of localized pgd, which makes sense. The authors should be more explicity
-evaluations are weak. They do not consider adaptive attacks and conduct mostly an initial investigation of the defense.
-Expand related work. This defense is similar to circuit-breakers for LLMS https://arxiv.org/pdf/2406.04313

All in all, this is a solid workshop paper. The paper is on theme for the workshop. Similar defenses have been proposed in other parts of the robustness literature, but I am not familiar with similar papers for patch attacks. The paper should be interesting to discuss in person at the workshop.